# Preventive Effect of Pharmaceutical Phytochemicals Targeting the Src Family of Protein Tyrosine Kinases and Aryl Hydrocarbon Receptor on Environmental Stress-Induced Skin Disease

**DOI:** 10.3390/ijms24065953

**Published:** 2023-03-21

**Authors:** So Jeong Paik, Dong Joon Kim, Sung Keun Jung

**Affiliations:** 1School of Food Science and Biotechnology, Kyungpook National University, Daegu 41566, Republic of Korea; sjpaik0916@gmail.com; 2Department of Microbiology, College of Medicine, Dankook University, Cheonan 31116, Republic of Korea; kjoon95@dankook.ac.kr; 3Research Institute of Tailored Food Technology, Kyungpook National University, Daegu 41566, Republic of Korea

**Keywords:** phytochemical, skin disease, prevention, src family of protein tyrosine kinases, aryl hydrocarbon receptor, environmental stress, signaling pathways

## Abstract

The skin protects our body; however, it is directly exposed to the environment and is stimulated by various external factors. Among the various environmental factors that can threaten skin health, the effects of ultraviolet (UV) and particulate matter (PM) are considered the most notable. Repetitive exposure to ultraviolet and particulate matter can cause chronic skin diseases such as skin inflammation, photoaging, and skin cancer. The abnormal activation of the Src family of protein tyrosine kinases (SFKs) and the aryl hydrocarbon receptor (AhR) in response to UV and/or PM exposure are involved in the development and aggravation of skin diseases. Phytochemicals, chemical compounds of natural plants, exert preventive effects on skin diseases through the regulation of various signaling pathways. Therefore, this review aims to highlight the efficacy of phytochemicals as potential nutraceuticals and pharmaceutical materials for the treatment of skin diseases, primarily by targeting SFK and AhR, and to explore the underlying mechanisms of action. Future studies are essential to validate the clinical potential for the prevention and treatment of skin diseases.

## 1. Introduction

Skin is the largest organ that covers the body and protects it from environmental stress [1]. It comprises two main layers, i.e., the dermis and epidermis, and keratinocytes constitute 95% of the latter [2]. Keratinocytes originate from epidermal stem cells in the basement membrane of the skin; after their development, they continue to move upward and are finally differentiated to form the stratum corneum, the main structure of the epidermis of the skin [3]. Keratinocytes in the stratum corneum follow a brick-and-mortar pattern of formation. This formation facilitates the prevention of various environmental damages caused by ultraviolet (UV) radiation, pathogenic microbiome, parasites, and surrounding pollutants.

UV is important to human beings for killing pathogens and vitamin D synthesis; however, acute and chronic exposure to UV rays promotes various skin diseases, including edema, erythema, wrinkles, and skin cancer [4]. Using various animal models, it has been demonstrated that skin inflammation [5,6], skin aging [7,8], and skin cancer [9,10] can be induced depending on the type and duration of UV rays. The International Agency for Research on Cancer under the World Health Organization found that diesel exhaust gas adversely affects human health; it was also identified as a first-class carcinogen [11]. Among various environmental pollutants, particulate matter (PM) is the major cause of skin damage [12]. Although particulate matter (PM) hardly penetrates healthy skin, it has been reported that contact with PM causes disease aggravation in people with damaged skin barriers, such as those with atopic dermatitis [13]. Therefore, PM is also recognized as one of the major environmental factors that cause skin damage.

The Src family of protein tyrosine kinases (SFKs) and the aryl hydrocarbon receptor (AhR) play important roles in maintaining homeostasis in various cells. However, hyperactivated SFK and AhR in response to environmental stressors, such as UV and PM, results in the activation of signal transduction related to the induction of various skin diseases [14,15]. Furthermore, SFKs and AhR are major causes responsible for the deterioration of skin health. A variety of external stimuli can stimulate SFKs and AhR, leading to the initiation of signaling cascades. After the activation of signaling responses to various external stimuli, skin diseases such as skin inflammation, psoriasis, atopic dermatitis, and skin cancer can occur [16,17,18,19,20,21]. Acute and chronic UV exposure in rodent skin causes abnormal phosphorylation of Fyn and Src, leading to skin inflammation, photoaging, and skin cancer [22]. In addition, PM directly binds to activated AhR and is subsequently involved in the onset of skin inflammation and skin carcinogenesis [23].

A nutraceutical is a health functional food or dietary supplement that is expected to help prevent or alleviate diseases in addition to the supplementation of basic nutritional components [24]. Phytochemicals are an important source of nutraceuticals and may effectively prevent skin diseases by maintaining skin homeostasis that can be imbalanced through environmental stress such as UV and PM [16,25]. Cellular signaling pathways are critical in maintaining homeostasis, and the hyperactivation of Src and the AhR leads to the onset of skin diseases [17,18,26,27]. Therefore, the consumption of phytochemicals could be a possible strategy to prevent skin diseases by blocking Src- and AhR-induced signaling pathways. The results of this study demonstrated that SFKs and AhR are major targets for preventing environmental stress-induced skin diseases and that pharmaceutical phytochemicals can be used to prevent skin disease through the regulation of SFKs and the AhR. The purpose of this study was to prove that phytochemicals can be effective in preventing skin diseases owing to their ability to control molecular mechanisms. The study results suggested that phytochemicals are a safe pharmaceutical material to maintain skin health in response to environmental pollution. 

## 2. Environmental Factors That Cause Skin Diseases

### 2.1. UV Irradiation in Skin Diseases

Although there are various environmental factors that can directly affect skin health, UV is the primary factor that disrupts human skin homeostasis. UV is a type of magnetic radiation, and the wavelength of UV is 10–400 nm, which is shorter than that of visible light but longer than that of X-rays. The electromagnetic spectrum of UV is divided into ultraviolet A (UVA), ultraviolet B (UVB), and ultraviolet C (UVC), with wavelengths of 315–400, 280–315, and 100–280 nm, respectively [28]. Most of the UVC used as a source of germicidal lamp is absorbed by the ozone layer in the atmosphere, and most of the UV range reaching the Earth’s surface is UVA [4]. UVB reaches the Earth’s surface at a maximum of 5% even in cloudless and sunny areas. It is considered a complete carcinogen because a single treatment causes skin cancer [29]. Although UVA causes relatively low damage to the skin, compared with UVB, repetitive exposure to UVA has also been reported to cause skin inflammation and skin cancer [30]. The effects of UV irradiation on mice are summarized in Table 1.

#### 2.1.1. Skin Inflammation

Acute UV irradiation causes DNA damage, including pyrimidine dimers and 6–4 photoproducts [33]. If left untreated by the DNA proofreading system of mammalian cells, it leads to aberrant gene expression. As with other inflammation, it is accompanied by redness, swelling, and pain, and these are representative symptoms of erythema. In previous in vivo studies, acute UVB exposure (0.5 J/cm^2^) resulted in skin inflammation and increased thickness of the epidermis in ICR and SKH-1 hairless mice [16,31]. 

#### 2.1.2. Skin Aging

Skin aging is categorized into intrinsic and extrinsic aging based on certain triggers. Unlike intrinsic aging caused by individual genetic characteristics, extrinsic aging is caused by environmental factors, mainly UV; thus, extrinsic aging and photoaging are considered the same [34]. It is generally known that repetitive exposure to UV rays results in the decomposition and collapse of collagen in the dermal layer of the skin, causing wrinkle formation, a phenomenon of photoaging [35]. In our previous study, a gradual increase in low-dose UVB irradiation appeared to cause skin photoaging in SKH-1 mice. One minimal erythema dose (MED) was defined as 45 mJ/cm^2^, and the irradiation dose for mice dorsal skin was increased weekly from 1 MED to 4 MED (0.18 mJ/cm^2^). Irradiation at 4 MED was maintained until 15 weeks. At the end of the experiment, wrinkle formation, increased epidermal thickness, and hyperplasia of keratinocytes were confirmed to occur on mouse skin [32].

#### 2.1.3. Skin Carcinogenesis

Skin carcinogenesis occurs in three stages: initiation, promotion, and progression [36]. In the initiation stage, acute UV irradiation causes prompt and irreversible DNA damage [37]. The promotion stage includes the clonal expansion of abnormal cell harboring DNA mutation [38]. Chemoprevention, prevention, and delay of carcinogenesis with the use of natural and synthetic medications are also considered promotion stages as they can be reversible with appropriate medication. However, once the neoplasm reaches the progression stage, it becomes difficult to control because cancer cells have a metastatic capacity [39]. Therefore, cancer prevention should take precedence over treatment. Because inflammation is considered a promotion stage in carcinogenesis, diseases such as cancer can be prevented through appropriate relief of inflammation. In a previous study, repetitive exposure to UV rays on the skin caused chronic inflammation and skin cancer in SKH-1 hairless mice. In addition, the pretreatment of natural flavonoid before UV exposure inhibited the development of skin tumors as well as the expression of inflammation-related biological markers [22]. Therefore, controlling inflammation to suppress the cancer progression may be an appropriate cancer prevention method.

### 2.2. Particulate Matter in Skin Diseases 

Air pollution has been reported as one of the causes of worsening skin diseases such as atopic dermatitis and extrinsic aging in several epidemiological studies. Vierkotter et al. suggested that air pollution correlated with a 20% increase in the presence of rough wrinkles and pigment spots on the forehead and cheeks [40]. A study by Hüls et al. clarified a relationship between pigment spots on the cheeks and NO_2_ [41]. A German cohort study found that the cumulative ozone exposure was correlated with the development of coarse facial wrinkles [42].

#### 2.2.1. Air Pollution 

Models of skin damage caused by PM exposure are being constructed; furthermore, standardized PM—a standard reference material available from National Institute of Standards and Technology—is used to establish PM-induced skin damage models. Various mechanisms of action have been reported regarding the toxicity of human-derived skin cells and rodents caused by PM, including the overproduction of reactive oxygen species (ROS) [43], the overexpression of inflammatory proteins such as cyclooxygenase (COX)-2 [44,45], and the overproduction of inflammatory cytokines [46,47].

#### 2.2.2. Cohort Analysis

A recent Cohort of Child Origin of Asthma and Allergic Disease study reported that exposure to fine dust in the early stage of pregnancy can cause fetal skin barrier dysfunction and increased atopic dermatitis induction at the age of 1 year [12]. In addition, a survey conducted on 4907 French children aged 9–11 years reported a correlation between PM10, NO_2_, NO_x_, and CO concentrations and eczema outbreaks [48]. In a study conducted on 7030 domestic children aged 6–13 years, a high correlation was observed between atopic dermatitis and parents’ smoking habits [49].

## 3. The Role of SFKs in Skin Disease

### 3.1. Structural Characteristics of SFKs

SFKs are important regulators in signaling transduction associated with various cell receptors’ response to the stimulation of environmental stress. Src, Fyn, Yes, Yrk, Blk, Fgr, Hck, Lck, and Lyn are included in SFK [14]. Among them, Src is particularly well known as a proto-oncogene that is closely associated with cancer development and progression in normal mammalian cells. SFKs play a significant role in oncogenesis through the mediation of multiple signaling cascades and are frequently highly activated in tumors [50]. Furthermore, SFKs share the same form of domain structure across all family members and are composed of four Src homology (SH) domains, which perform different functions [51]. Similar to kinases, SFKs have N-terminus and C-terminus at both ends of their structure, including the SH4, SH3, SH2, and SH1 domains. The common structure of SFKs is presented in Figure 1.

The structure of SFK is divided into two predominant domains as follows: stability-related domains and kinase activity-related domains. The SH4 domain is a region located at the N-terminus of SFK and can be myristoylated and/or palmitoylated. Myristoylation and palmitoylation may contribute to the stabilization of the protein structure. Myristoylation is included in the structure of all SFKs whereas palmitoylation is not. Myristoylation is a reaction that adds myristate that consists of 14-carbon fatty acid to the N-terminal glycine residue of SFKs and leads to the formation of a covalent amide bond. Palmitoylation is an addition reaction that adds palmitate to a Cys residue of a protein and leads to the formation of stable amide linkage [52]. Myristoylation and palmitoylation are the main types of protein lipidation, a process in which various lipid and lipid metabolites are produced for transforming proteins in eukaryotic cells. If these are preceded, membrane localization may be possible, and membrane binding becomes stronger than before. Thus, these processes may lead to the regulation of protein trafficking, stability, and aggregation [52,53,54,55].

The SH3 domain consists of approximately 50 amino acids, and this domain is one of the peptide recognition modules; it is related to various signaling pathways, such as cell growth control, cytoskeleton organization, and endocytosis. This domain is important for protein–protein interactions, which may be caused by the assembly of specific protein complexes. The SH3 domain and its specific ligand form specific protein complexes, and its ligand recognition can occur through interaction with polyproline and proline-rich (PxxP) binding sites [56,57].

The SH2 domain specifically combines with phosphorylated tyrosine containing peptide segments. This is a key element in the regulation of tyrosine kinase signaling and involves connecting proteins to signaling systems initiated by tyrosine phosphorylation [56]. Through these processes, the SH2 domain may affect the SFK activities. Aside from the SH2 domain, the SH3 domain and C-terminal are also associated with SFK activities [14].

The SH1 domain is the most preserved among all SFKs, and this domain is located at the C-terminal of the kinase. The SH1 domain is a kinase active site that contains tyrosine phosphorylation activation such as Tyr416 (Tyr419 in human) residue and an ATP-binding pocket. The autophosphorylation of Tyr416 residue is associated with cancer development, whereas Tyr527 phosphorylation prevents cancer. Phosphorylated Tyr527 performs intramolecular interactions with the SH2 domain of SFKs, leading to intramolecular SH3 domain-mediated interactions that inhibit catalytic activity [58]. Thus, the SH1 domain is important as it is closely related to carcinogenesis [53].

### 3.2. The Role of SFKs in Skin Diseases

#### 3.2.1. Upstream and Downstream Regulators for SFK Activity

SFK activity is closely related to upstream and downstream regulators of SFKs. For Src, Src activation can be determined through C-terminal Src kinase (CSK) and Src autophosphorylation [50]. CSK homologous kinase phosphorylates Tyr527 residue at the C-terminal of the kinase. When phosphorylated, Tyr527 binds to the SH2 domain, the C-terminal, SH2, and SH3 domains of Src facilitate intramolecular interaction [58]. The SH1 domain cannot facilitate kinase activity with this interaction; thus, Src is converted to an inactive form that is not involved in carcinogenesis [59]. Receptor-type protein tyrosine phosphatase α (RPTPα), dephosphorylates Tyr527, and may cause conformational change in intramolecular interaction. Contrarily, the Tyr416 residue in the SH1 domain is autophosphorylated [60]. Src is fully activated by the autophosphorylation of Tyr416 at the activation loop [61]. Therefore, Tyr416 phosphorylation induces tumorigenesis as opposed to Tyr527.

Src plays important roles by extensively interacting with transmembrane receptor tyrosine kinases (RTKs) through the SH2 and SH3 domains [62]. Epidermal growth factor receptor (EGFR), human epidermal growth factor receptor 2 (HER2/ErbB2), platelet-derived growth factor receptor, insulin-like growth factor-1 receptor (IGF-1R), and c-Met/hepatocyte growth factor receptor are included in RTKs, and RTKs interact with Src and convey the phosphate. After the interaction, Src proceeds as cascades regulated by RTKs and directly transduces signals to its downstream factors. Src has been found to affect phosphoinositide 3-kinases (PI3Ks)/Akt and the signaling transducer and activator of transcription 3 (STAT3). Other receptors located in the membrane, such as integrins and erythropoietin receptor (EpoR), can be related to Src activation [50].

Similarly, other SFKs can be phosphorylated by CSK or be autophosphorylated in specific residues. The activated SFK functions by transmitting signals to the downstream factors [63,64].

#### 3.2.2. Importance of SFK Activities in Skin Diseases

SFK activation induces signaling that leads to various biological functions, and the participation of other cell receptors is also associated with functional determination [59]. Among SFKs, it is well known that Src is closely related to the development of many skin diseases, such as skin inflammation and skin cancer [16,22,65,66]. 

Many experimental methods have been developed to prove that Src is the main factor affecting skin diseases. Knockdown of the Src gene, the use of Src inhibitors, and the use of Src-related gene knockout mice can be presented as effective ways to determine whether Src plays a crucial role in pathogenesis. 

Transfection of HaCaT cells using siRNA targeting Src was performed to demonstrate that Src is a main factor affecting skin inflammation, such as COX-2 expression and mitogen-activated protein kinase (MAPK) phosphorylation. Knockdown of Src mRNA resulted in decreased COX-2 overexpression and blocked MAPK phosphorylation induced by UV irradiation. It also inhibited the phosphorylation of the Tyr845 residue of EGFR known to interact with Src [16]. Src inhibitors can be used for blocking Src activation. As an SFK inhibitor, PP2 (AG 1879) can be treated to the HCT 116 human colon cancer cell lines to confirm the role of Src in wound healing. As a result, the inhibition of Src activity led to a reduction in the phosphorylation of Src Tyr416 and EGFR Tyr845 and affected the downstream factors that facilitate wound healing [67]. Knockout (KO) mice can be used to show that CSK plays a key role in anti-cancer effects. CSK is known to activate Src via phosphorylation of the Tyr527 residue related to anti-cancer effects. In the epidermis of K5-CSK KO mice, Tyr529 (same as Tyr527) phosphorylation was decreased, whereas Tyr418 (same as Tyr416) phosphorylation was increased. In addition, CSK KO mice exhibited apparent epidermal hyperplasia. As the age of the mice increased, hyperplasia spread to the entire skin, and skin thickness also increased. Chronic inflammation was observed in these mice. This evidence showed that Src and CSK play an important role in the regulation of skin inflammation and cancer development [68]. Therefore, targeting SFKs, including Src, can be a powerful method for preventing skin diseases via the regulation of various signaling pathways.

#### 3.2.3. Molecular Mechanism of Src in Skin Diseases

Various factors can activate SFKs, and increased SFKs activity causes inflammation and cancer. Inflammation is a self-limiting reaction that produces anti-inflammatory cytokines against the production of proinflammatory cytokines. However, chronic inflammation can occur if inflammatory factors persist or if the mechanisms required to complete the inflammatory response fail. Chronic inflammation is known to be crucial in malignant tumor development; thus, inflammation and cancer are closely related [69].

Among SFKs, Src has been proven to be an important kinase that induces an inflammatory response due to phosphorylation [67]. Activated Src interacts with EGFR, and EGFR is activated. Activated EGFRs form dimer, and the EGFR dimers stimulate many intracellular signaling pathways [68]. Activated EGFR can stimulate downstream pathways such as the MAPK and phosphatidylinositol 3-kinase (PI3K)/Akt signaling pathways [70].

MAPK cascades are composed of MAPK kinase kinases (MAP3Ks), MAPK kinases (MAP2Ks), and MAPK-linked phosphorylation kinases. MAP3K can be activated by stimuli such as environmental stress, cytokines, and growth factors [71]. Especially, it is well known that UVB [72] and PM can, as environmental factors, mediate MAPK signaling [12]. These stimuli activate MAP3Ks, including Raf, mixed-lineage kinase, and MAPK/extracellular signal-regulated kinase (ERK) kinase kinases, and MAP3Ks deliver phosphate groups to its downstream MAP2Ks. MAPK kinases and PAK1 are MAP2K components. Phosphorylated MAP2K also phosphorylates its downstream MAPK. ERK, Jun N-terminal kinase, and p38 are also MAPK components. Phosphorylated MAPK affects the transcription factors that exist in the cell nucleus [71,73].

The PI3K/Akt signaling pathway consists of a connection between PI3K and Akt. Extracellular stimuli can stimulate PI3K, and then PI3K forms heterodimers composed of the p85 and p110 subunits for stabilization. p85 is an inhibitory adaptor/regulatory subunit. It incorporates signals from intracellular proteins, such as Src, Ras, and protein kinase C (PKC), and RTKs. p85 then offers integration points for the activation of catalytic subunit p110 and its downstream molecules [74,75]. Activated PI3K converts phosphatidylinositol-4,5-bisphosphate (PIP2) into phosphatidylinositol-3,4,5-triphosphate (PIP3) by adding a phosphate group to PIP2. Contrarily, PIP3 may also be dephosphorylated by a negative regulator of PI3K signaling, PTEN, which can be reversed to PIP2. Akt is a downstream of PI3K and binds to PIP3 at the cell membrane. Phosphoinositide-dependent kinase 1 (PDK1) also binds to PIP3 and phosphorylates threonine 308, a kinase domain of Akt, which contributes to the stabilization of the activation loop [74,75,76].

Signaling cascades activate nuclear factor kappa B (NF-κB) or upregulate activator protein (AP)-1 transcription factors that promote inflammation and carcinogenesis. The expression of these transcription factors activates several inflammatory genes encoding tumor necrosis factor (TNF)-α, COX-2, and inducible nitric oxide synthase (iNOS) and subsequently produces TNF-α, prostaglandin E_2_ (PGE_2_), and nitric oxide (NO), respectively [10,70,77,78]. These factors can indicate skin diseases [77,79]. COX-2 promotes disease progression ranging from inflammation to cancer; thus, COX-2 is a disease-inducible enzyme [80]. It is also considered a crucial factor that represents skin inflammation and skin carcinogenesis [22]. TNF-α and iNOS can be accompanied by skin inflammatory reactions [70]. Therefore, blocking Src activation is important to exert anti-inflammatory and anti-cancer effects. Src-mediated signaling pathways are presented in Figure 2.

## 4. Molecular Mechanism of AhR in Skin Diseases 

AhR is a transcription factor that is influenced by different environmental stimuli, and their exposure initiates signaling cascades related to inflammation and cancer [81,82]. Interestingly, AhR signaling paradoxically functions as a treatment for skin barrier dysfunction; however, at the same time, it exacerbates skin diseases through ROS generation [19,27].

### 4.1. AhR Ligands 

AhR is located in the cytoplasm, and some low-molecular-weight chemicals are AhR ligands that can promote AhR activation [83]. Ligands with a proper size of three benzene rings are known to bind to AhR. There are several extrinsic substances known as AhR ligands. Extrinsic composites such as toxic pollutant polycyclic aromatic hydrocarbons and 2,3,7,8-tetrachlorodibenzo-p-dioxin (TCDD) are well-known AhR ligands. Extrinsic natural compounds, such as flavonoids, glucosinolates, and indocarbionols, and endogenous AhR ligands that form in the body, such as a tryptophan dimer 6-formylindolo[3,2-*b*]carbazole (FICZ), can also act as AhR ligands [84,85]. Furthermore, PM activates inflammatory responses in human skin by easily penetrating the skin layer [21]. Therefore, various external factors including environmental pollutants may activate AhR.

### 4.2. Structure and Molecular Mechanism of AhR

AhR is a multi-component complex composed of immunophilin-like Ah receptor-interacting protein (AIP, the same as hepatitis B virus X-associated protein 2 [XAP2]), two molecules of 90-kD heat shock protein (HSP90), HSP90 co-chaperone p23, and Src under a normal condition [82,86,87]. Hsp90 and AIP block AhR degradation, and AhR can translocate from the cytoplasm to the nucleus. Then, DNA binding occurs from its translocation [15,82,88].

Translocated AhR combines with the aryl hydrocarbon receptor nuclear translocator (ARNT). AhR and ARNT forms heterodimer, and its C-terminal acts with its promoter, known as aryl hydrocarbon response elements (same as xenobiotic response elements [XREs], 5′-(C)GCGTG-3′). AhR–ARNT recruit coactivator and transcript genes related to xenobiotic metabolism, including cytochrome P450 family 1 subfamily A member 1 (CYP1A1), cytochrome P450 family 1 subfamily B member 1 (CYP1B1), and AhR repressor (AhRR) [15,82,89]. CYP1B1 and CYP1B1 metabolite AhR ligands to mutagenic epoxide intermediates that may lead to ROS generation; then, inflammation may occur due to the ROS production [82,90].

On the other hand, the promotion of the CYP1A1 gene’s expression ironically strengthens the disrupted skin barrier [19]. The AhRR disrupts AhR–ARNT complexes, suppressing inflammatory responses and carcinogenesis induced by AhR signaling [91].

Additionally, the AhR ligand binds to AhR, and the resulting AhR complex releases Src; then, Src interacts with EGFR. Activated EGFR mediates several cellular events, such as MAPK signaling [89,92], and can also produce inflammatory responses, such as COX-2 expression [73,93].

Activated AhR is well known to be closely associated with skin inflammation and skin carcinogenesis [89]. UV radiation produces an AhR ligand FICZ, and exposure to UVB has been demonstrated to mediate AhR signaling [94,95]. PM also affects the activation of AhR and contributes to the outbreak of skin inflammation [96]. Therefore, targeting AhR can be an effective way to treat environmental stress-induced skin diseases at the molecular level.

AhR-mediated signaling pathways affected by various stimuli are summarized in Figure 3.

## 5. Prevention of Skin Diseases Using SFKs and/or AhR-Targeted Phytochemicals

Targeting SFKs and/or AhR is an effective way to prevent skin diseases, including inflammation and skin carcinogenesis caused by different signaling cascades, as described above. It has been demonstrated that phytochemicals prevent skin diseases caused by environmental factors such as UVB [9,16,17,26,27,97]. Therefore, research into phytochemicals targeting SFKs and/or AhR may be a key strategy for alleviating skin diseases.

### 5.1. Functions of SFK-Targeted Phytochemicals in Skin Diseases

The molecular functions of SFK-targeted phytochemicals in skin diseases are summarized in Table 2 and presented schematically in Figure 4.

4-Phenylpyridine (4-PP) suppressed skin inflammation caused by UVB by targeting c-Src in pre-clinical studies. 4-PP decreased inflammation markers, such as COX-2, AP-1, and PGE_2_ expression, by blocking c-Src-mediated MAPK phosphorylation in human immortalized cell lines. 4-PP reduced epidermal thickness increased by UVB exposure and blocked COX-2 expression, resulting in the phosphorylation of EGFR in UVB-irradiated mice [16].

*Datura metel* L. contains total withanolides; these phytochemicals can relieve psoriasis-like skin inflammation induced via IMQ application. IMQ is a potent immune activator that stimulates Toll-like receptor (TLR)7/8; thus, it can exaggerate chronic inflammatory skin diseases such as psoriasis. Total withanolides restored IMQ-induced skin damage in male BALB/c mice through various pathways. Furthermore, the phosphorylation of STATs and MAPKs induced by cytokines was decreased, as well as the phosphorylation of Lck, Lyn, and Src [103].

Malic acid and isocitric acid improved wound healing by targeting c-Src. The phosphorylation of Src Y416, EGFR (upstream of Src), and STAT3 Y705 (downstream of Src) was blocked by the treatment of malic acid and isocitric acid [67].

Danshensu inhibited abnormal epidermal proliferation through the yes-associated protein (YAP) pathway, resulting in improved psoriasis. Src has been proven to drive tumor development and metastasis through the activation of the YAP/PD-binding motif (TAZ) axis [104] and lead to the expression of COX-2 [105]. Danshensu was found to suppress M5-induced cell apoptosis and regulate cell cycle-related factor expression in HaCaT cells. Furthermore, it decreased YAP mRNA and protein expression in both HaCaT cells and BALB/c mice.

Myricetin inhibited UVB-induced skin carcinogenesis by targeting Fyn kinase. It regulated Fyn downstream pathways and bound directly to Fyn to block Fyn kinase activity in JB6P+ mouse skin epidermal cells [9,98].

Quercetin exists in many natural plants and is a phyto-compound that downregulates Src expression levels in human keratinocytes. Thus, psoriasis, which frequently occurs along with skin inflammation, can be alleviated by the action of quercetin [99].

Caffeic acid generally found in coffee blocks elevated Fyn activity. Caffeic acid also decreased COX-2 expression and its promoter activity by directly binding with Fyn kinase. Therefore, caffeic acid can prevent UVB-induced skin carcinogenesis [100].

Cryptotanshinone, a component of *Salvia miltiorrhiza Bunge*, is effective in improving atopic dermatitis. It prevented Lyn and Syk phosphorylation and blocked the MAPK and NF- κB signaling pathways. It also blocked phospholipase C (PLC)γ and PKCδ phosphorylation in cells. The topical treatment of cryptotanshinone improved DNCB-induced atopic dermatitis-like skin lesions in BALB/c mice and decreased immunoglobulin E (IgE) production in mice serum and mice ear [101].

The combination of ursolic acid and curcumin can prevent TPA-induced skin carcinogenesis by targeting c-Src. It is reported that TPA induces skin diseases such as skin cancer [97], oxidative stress, and inflammation [106]. TPA-treated mouse skin showed increased tumor multiplicity and higher incidence and latency of skin cancer; however, topical treatment with a combination of ursolic acid and curcumin reduced these phenomena. It was shown that the combination of ursolic acid and curcumin blocked the phosphorylation of EGFR Y1086 and Src Y416 residues in mice tissues [97].

Baicalein alleviates BaP-induced oxidative stress by regulating Src phosphorylation in human epidermal keratinocytes. BaP is a known environmental pollutant; the compound baicalein is a component of *Scutellaria baicalensis* root. Treatment with baicalein decreased ROS production and prevented the phosphorylation of Src caused by BaP treatment [102].

5-Deoxykaempferol exerts efficacy on UVB-induced skin cancer by targeting the Src and PI3K pathways. Src was a main target of 5-deoxykaempferol, and the blocking of Src activation decreased the phosphorylation of its downstream factors. A pull-down assay showed the direct binding of 5-deoxykaempferol to Src, PI3K, and RSK in JB5 P+ mouse epidermises [26].

The above literature shows that phytochemicals from natural plants can prevent skin diseases by modulating intracellular signal cascades induced by various stimuli. However, studies on the effect of phytochemicals on air pollutant-induced skin diseases have not been actively investigated except for BaP. Therefore, we propose that it is necessary to prove that phytochemicals can prevent the exacerbation of skin conditions caused by atmospheric pollutants such as PM.

### 5.2. Phytochemicals Targeting AhR for the Regulation of Skin Diseases

AhR is a double-edged sword because it has two opposite effects on the skin: improving atopic dermatitis and worsening skin inflammation. Atopic dermatitis is closely associated with skin barrier function, and the upregulation of keratinocytes differentiation is important when forming a skin barrier [107]. The activation of AhR leads to the expression of the CYP1A1 gene, and CYP1A1 gene expression also upregulates skin barrier formation [19]. AhR is a positive upstream of skin barrier proteins; thus, activating AhR is important for alleviating atopic dermatitis [108].

Interestingly, AhR is a negative upstream of skin inflammation caused by UVB. Elevated CYP1A1 expression produces ROS and causes skin inflammation by mediating oxidative stress on the skin [27]. Therefore, research on the CYP1A1 gene that affects various functions of the human skin is necessary.

The molecular functions of AhR-targeted phytochemicals in skin diseases are summarized in Table 3 and presented schematically in Figure 5.

Diosmin, a flavone glycoside, belongs to the Rutaceae family. Diosmin activates AhR signaling pathways and alleviates atopic dermatitis. When treated with diosmin, the expression of CYP1A1 gene, a AhR downstream was increased, and the expression of differentiation markers, such as filaggrin (FLG), loricrin (LOR), hornerin (HORN), involucrin (IVL), and ovo-like transcriptional repressor 1 (OVOL1) are also increased. It has been demonstrated that diosmin directly binds with AhR and activates AhR downstream. As a result, diosmin improves atopic dermatitis by restoring skin barrier function via the AhR signaling pathways [19].

Cinnamaldehyde is the main component of *Cinnamomum cassia*. Cinnamaldehyde reduced BaP-induced oxidative stress through the AhR pathway. BaP activates the AhR pathway and leads to the expression of CYP1A1 gene. In addition, increased CYP1A1 gene expression induces ROS production and oxidative stress on HaCaT cells. However, cinnamaldehyde decreased BaP-induced CYP1A1 expression and its upstream, AhR. Regulated AhR blocked the translocation from the cytosol to the nucleus [20]. Additionally, cinnamaldehyde decreased the oxidative stress production via the NRF2/HO-1 pathway known as an ROS regulator [112]. As a result, cinnamaldehyde reduced ROS production and had an antioxidant effect on the skin [20].

Cynaropicrin is a chemical found in artichoke (*Cynara scolymus*) that has antioxidant effects. Without UVB irradiation, cynaropicrin translocated AhR from the cytoplasm to the nucleus, leading to the mRNA expression of its downstreams, namely, CYP1A1, NRF2, and Nqo1. Cynaropicrin also reduced oxidative stress induced by UVB irradiation. Cynaropicrin treatment translocated NRF2 to the nucleus in response to UVB-induced oxidative stress. As a result, inflammatory responses such as ROS, IL-6, and TNF- α production has decreased in human keratinocytes. Therefore, it has been demonstrated that cynaropicrin attenuates oxidative stress induced by UVB via the AhR-NRF2-Nqo1 signaling pathway [27].

Baicalein is a potent inhibitor of BaP-induced oxidative stress. BaP phosphorylates Src and translocates AhR to nuclear. AhR located in the nucleus increases the expression of CYP1A1; the activation of AhR-CYP1A1 and the subsequent ROS production are known to induce proinflammatory cytokines secretion. However, baicalein inhibited AhR nuclear translocation and reduced CYP1A1 expression in human keratinocytes. In addition, baicalein activates the NRF2-HMOX1 pathway. Consequently, baicalein reduced oxidative stress through the inhibition of ROS and proinflammatory cytokine production [102].

Kaempferol prevents UVB- and AhR-mediated gene expression. Transcriptomics has proved that kaempferol treatment regulates gene expressions related to AhR, NRF2, and NF-κB. Kaempferol also blocks NF-κB p65 and RelB nuclear translocation in HaCaT and NHEK cells, indicating its anti-inflammatory effects on the skin [109].

Pterostilbene prevents PM-induced skin disease by targeting AhR. PM can aggregate skin inflammation and skin aging as well as reduce the moisturizing ability of the skin by activating AhR. Pterostilbene decreased the nuclear translocation of AhR and downstream gene CYP1A1 expression, as well as COX-2 expression and MAPK phosphorylation. The expressions of matrix metalloproteinase (MMP)-1, MMP-2, and MMP-9 that represent skin aging were also decreased, whereas moisturizing protein aquaporin-3 (AQP-3) was increased in HaCaT cells and CCD-966SK dermal fibroblasts. This indicates that pterostilbene alleviates skin inflammation, aging, and oxidative stress induced by PM by blocking AhR activation [110].

Oleanolic acid has been proven to prevent skin aging caused by PM. It decreases the expressions of CYP1A1, TNF-α, and IL-6, as well as the expressions of MMP-1 and autophagy related proteins p62 and LC-3 I and II in human epidermal keratinocytes and dermal fibroblasts. This indicates the therapeutic effects of oleanolic acid on PM-induced skin diseases by targeting AhR signaling [111].

The evidence in these studies shows that phytochemicals can regulate AhR activity in different directions, depending on the target skin disease. However, phytochemicals have only been shown to be able to prevent skin diseases by regulating AhR-related signaling cascades at the cellular level. Existing research has not proven the preventive effects of phytochemicals in vivo. Therefore, further research is needed to clarify the mechanism of the action of phytochemicals in animal studies.

## 6. Limitations of Animal Model for AhR Targeting Phytochemicals

It is important to prove that phytochemicals can improve skin health by controlling AhR activity. However, it has not been demonstrated that phytochemicals affect AhR activation in vivo. Therefore, we investigated which in vivo markers could prove that AhR is an essential factor in maintaining skin health. The experimental mouse models of AhR-related skin diseases are summarized in Table 4.

There are several ways to demonstrate that AhR is a key regulator of skin disorders.

The silencing of the AhR gene in mice can be used to prove that AhR is the main factor in maintaining skin health. Yu et al. used AhR-null mice and induced atopic dermatitis in mice skin using MC903. MC903-treated wild-type mice and AhR-null mice had differences in ear thickness and scratching frequency. In AhR-null mine skin, ear thickness and scratching frequency were decreased [113]. Smith et al. proved that tapinarof alleviates skin inflammation through AhR. Tapinarof is a compound that is derived from bacteria. IMQ-treated AhR-sufficient and AhR-knockout mice were topically administered tapinarof, and their clinical scores and cytokine transcription levels were compared. IMQ-induced clinical score increased and elevated IL-17A and IL-17F levels were decreased when treated with tapinarof in AhR+/+ mice; however, there was no difference in AhR−/− mice. This indicates that tapinarof attenuates psoriasis and atopic dermatitis via activation of AhR [114]. To clarify whether AhR is essential for maintaining skin barrier integrity in mice, AhR KO mice also can be used. Increased TEWL, structural changes in epidermis, and microarray results show that AhR is a key factor in skin barrier function [117].

AhR antagonists can be used to demonstrate the effectiveness of substances against skin diseases. Yu et al. used CH223191, an AhR antagonist, to prove the effect of tryptophan metabolites on atopic dermatitis. MC903 was used to induce atopic dermatitis and tryptophan metabolites, and CH223191 was also used to treat wild-type C57BL/6 and BALB/c mice. Tryptophan metabolites attenuated MC903-induced skin inflammation; however, CH223191-treated mice showed increased ear thickness, scratching frequency, TSLP mRNA level, and serum IgE level compared to CH223191+tryptophan metabolites-treated mice. This showed that AhR is required for the action of tryptophan metabolites on atopic dermatitis [113]. In addition, CH223191 can be used to show that esomeprazole inhibits fibro-inflammation in a BLM-induced BALB/c mice model. CH223191-treated mice showed increased dermal thickness and elevated mRNA levels of profibrotic markers such as COL1A1, α-SMA, CTGF, and TGF-β1 [116].

AhR agonists can be used to clarify the effect of AhR activation on skin health. Benvitimod, an AhR agonist, activates AhR, leading to alleviation of rosacea-like eruptions in female BALB/c mice skin. LL-37 was administered to induce rosacea-like eruption in mice; however, topical treatment of benvitimod decreased redness scores, redness areas, the infiltration of dermal inflammatory cells, TLR2 expression, and chemokine expression. Therefore, it is shown that AhR activation can be a positive target for treating rosacea [115].

The removal of AhR ligands from the diet impairs skin barrier integrity in C57BL/6 mice. AhR ligands bind with AhR, leading to AhR pathway activation. Activated AhR is related to skin barrier function; therefore, researchers removed AhR ligands from the mouse diet. Skin deterioration was observed in mice that on a diet without AhR ligands [117].

AhR-CA transgenic mice were used to show that the consecutive activation of the AhR pathway promotes skin inflammation. AhR-CA transgenic mice showed increased skin inflammation-related markers such as increased itching frequency, inflammatory gene expression, and cell-mediated immune response. The results show that the activation of AhR can aggravate skin inflammation in vivo [118].

As the homeostasis of AhR activity is important for maintaining human skin health, it is essential to identify the regulatory molecular mechanism of AhR activity using phytochemicals.

## 7. Conclusions and Future Research Directions

External environmental factors have recently emerged as major concerns that threaten human health. Since environmental stress can cause skin diseases, it is necessary to develop materials that can prevent diseases caused by external stimuli. This study demonstrated that environmental factors such as UV and/or PM activate various molecular targets including SFK and AhR and cause skin inflammation, photoaging, and skin cancer. Furthermore, it was confirmed that phytochemicals directly or indirectly suppress SFK or AhR activity to prevent the occurrence of skin diseases. Therefore, we suggest that the regulation of SFK and AhR signaling pathways using phytochemicals can help prevent environmental stress-induced skin diseases.

Phytochemical research has been widely described in therapeutic and preventive research; however, several studies have been identified that focus on the effects of a broad range of SFK and AhR-related signaling pathways. For phytochemical materials to be clinically researched and industrialized, it is thought that more in-depth research results on the identification and binding structure of the proteins to which they bind should be provided. Since the skin is the main target organ, most studies have focused on topical treatment; therefore, the degree of application to cosmeceutical is high. However, research on pharmaceutical applicability through oral or intraperitoneal administration is also considered necessary. It is widely described as a skin disease; however, as the environment changes, such as through global warming, the symptoms and causes appear in various ways. Therefore, future research on phytochemicals on various skin diseases such as atopy and psoriasis would also be of interest.

## Figures and Tables

**Figure 1 ijms-24-05953-f001:**
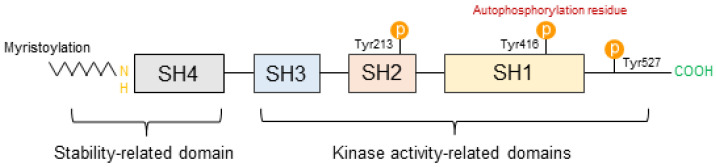
Structure of SFKs.

**Figure 2 ijms-24-05953-f002:**
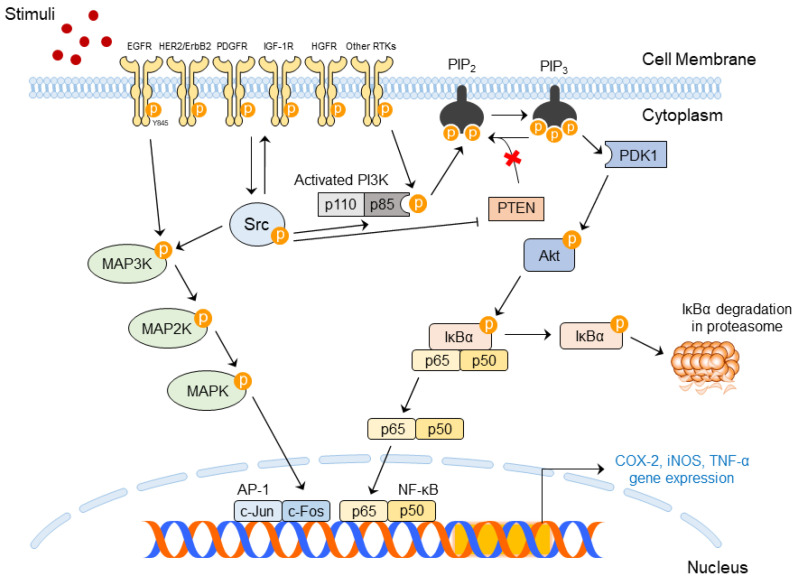
Src signaling pathways in skin inflammation and skin cancer.

**Figure 3 ijms-24-05953-f003:**
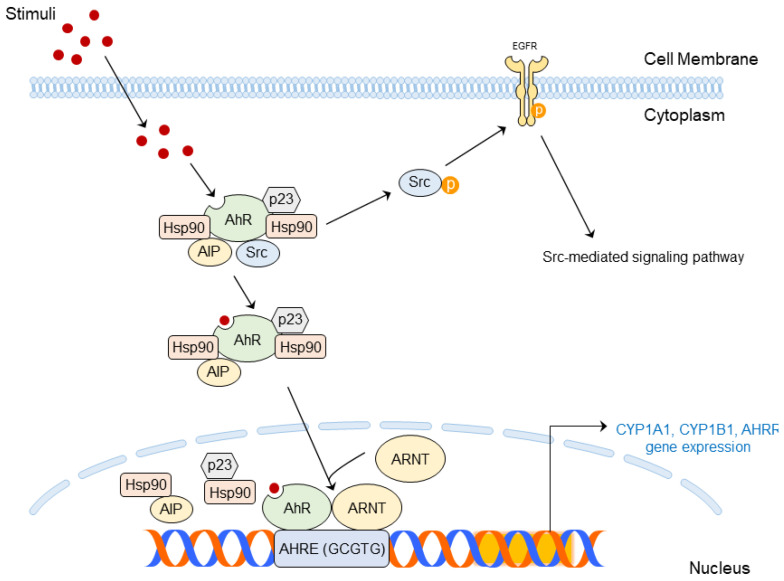
AhR-mediated signaling pathways in skin inflammation and skin cancer.

**Figure 4 ijms-24-05953-f004:**
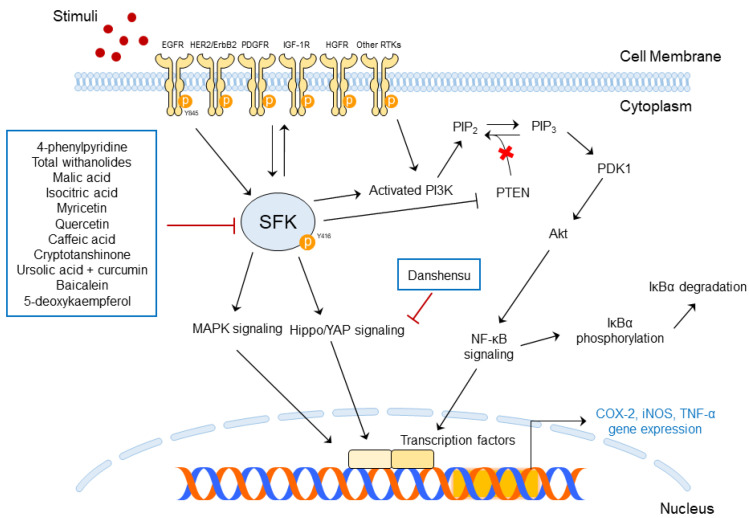
Regulation of SFK signaling pathways by phytochemicals in skin diseases.

**Figure 5 ijms-24-05953-f005:**
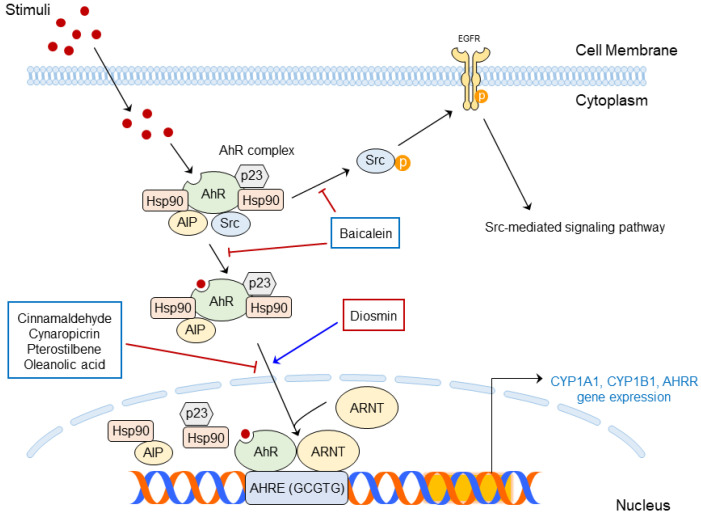
Phytochemicals targeting AhR signaling pathways for skin diseases.

**Table 1 ijms-24-05953-t001:** UV-induced skin diseases in mouse models.

UV Dose	Term	Rodent Species	Skin Diseases	Abnormal Changes	Ref
0.5 J/cm^2^	Single	ICR and SKH-1 hairless mice	Inflammation	Increased epidermal thickness	[16,31]
0.045–0.18 J/cm^2^	15 weeks (3 times/week)	SKH-1 hairless mice	Photoaging	Wrinkle formation, increase in epidermal thickness, irregular brown spot	[32]
0.18 J/cm^2^	15 weeks (3 times/week)	SKH-1 hairless mice	Skin cancer	Development of neoplasm, increase in epidermal thickness	[22]

**Table 2 ijms-24-05953-t002:** Molecular functions of SFK-targeted phytochemicals in skin diseases.

Name	Stimuli	Target Disease	Target Signaling	In Vitro Markers	In Vivo Markers	Ref
4-Phenylpyridine	UVB	Skin inflammation	Src	COX-2 PGE2 Direct binding with c-Src	Epidermal thickness	[16]
Total withanolides	Imiquimod (IMQ)	Psoriasis-like skin inflammation	Lyn Lck Src Treg/Th17 axis		Skin damage site pigmentation PASI score Dermatological changes	[17]
Malic acid		Wound healing	Src	EGFR Y845 phosphorylation Src Y416 phosphorylation STAT3 Y705 phosphorylation		[67]
Isocitric acid		Wound healing	Src	EGFR Y845 phosphorylation Src Y416 phosphorylation STAT3 Y705 phosphorylation		[67]
Danshensu	M5 (in vitro) IMQ (in vivo)	Psoriasis	YAP	YAP mRNA expression YAP expression	Gross morphology Epidermal thickening YAP expression	[18]
Myricetin	UVB	Skin cancer	Fyn	COX-2 AP-1, NF-κB transactivation MAPK phosphorylation P90RSK, MSK phosphorylation MEK, Raf, Fyn phosphorylation Fyn kinase activity Direct binding with Fyn	Fyn kinase activity COX-2 MAPK phosphorylation Direct binding with Fyn External appearance of tumors Number of tumors Tumor volume	[9,98]
Quercetin		Skin inflammation (psoriasis)	Src	Src, Lyn, Fyn expressions		[99]
Caffeic acid	UVB	Skin cancer	Fyn	COX-2 PGE2 generation Fyn kinase activity Direct binding with Fyn	COX-2 Fyn kinase activity Direct binding with Fyn	[100]
Cryptotanshinone	IgE, DNCB	Atopic dermatitis	Lyn, Syk, PLC	Lyn, Syk phosphorylation PLCγ, PKCδ, IKKβ phosphorylation	Skin pigmentation Ear thickness Accumulation of CD11b+ cells IgE in serum and ear p65 phosphorylation	[101]
Ursolic acid + Curcumin	12-ο-Tetradecanoylphobol-13-acetate (TPA)	Skin carcinogenesis	Src	EGFR Y1086, Src Y416, p70S6K T389, c-Jun S73 phosphorylation		[97]
Baicalein	Benzo[a]pyrene (BAP)	Oxidative stress	Src	CYP1A1 mRNA Src phosphorylation AhR translocation		[102]
5-Deoxykaempferol	UVB	Skin carcinogenesis	Src, Rsk	COX-2 Src phosphorylation Direct binding with Src, PI3K, RSK	Skin pigmentation Number of tumors Number of tumor-bearing mice Tumor volume Blood vessel formation	[26]

**Table 3 ijms-24-05953-t003:** Molecular functions of AhR-targeted phytochemicals in skin diseases.

Name	Stimuli	Target Disease	Target Signaling	In Vitro Markers	In Vivo Markers	Ref
Diosmin	AhR ligand (endogenous for kynurenine, exogenous for TCDD)	Atopic dermatitis	AhR	CYP1A1 transactivity↑ FLG, CYP1A1, NQO1, OVOL1, LOR, HRNR, IVL mRNA expression↑ proFLG, LOR, OVOL1, FLG, HRNR, IVL, NQO1 expression↑ Epidermal thickness↑ (3D-skin model) STAT3 phosphorylation↓ AhR nuclear translocation↑ Direct binding with AhR		[19]
Cinnamaldehyde	BaP	Oxidative stress	AhR NRF2/HO-1 pathway	CYP1A1 mRNA expression↓ AhR nuclear translocation X AhR expression↓ NRF2 expression↑ NRF2 nuclear translocation↑ HO-1 mRNA expression↑ ROS production↓		[20]
Cynaropicrin	UVB	Oxidative stress	AhR-NRF2-Nqo1 pathway	Without UVB radiation (1) AhR nuclear translocation↑ (2) CYP1A1, NRF2, Nqo1 mRNA expressions↑ (3) NRF2 nuclear translocation↑ With UVB radiation (1) ROS production↓ (in a NRF2 dose-dependent manner) (2) IL-6, TNF-α production↓		[27]
Baicalein	BaP	Oxidative stress	AhR NRF2/HMOX1 axis	CYP1A1 mRNA↓ HMOX1 mRNA↑ NRF2 nuclear translocation↑ ROS production↓ IL-1α, IL-1β↓ Src phosphorylation↓ AhR nuclear translocation↓		[102]
Kaempferol	UVB	X	AhR	Microarray p65, RelB translocation↓ PPARα, PPARβ transcriptional activity↑		[109]
Pterostilbene	PM	Skin inflammation Skin aging Decreased moisturizing ability	AhR	ROS generation↓ AhR translocation↓ CYP1A1 expression↓ MAPK phosphorylation↓ MMP-1, MMP-2, MMP-9↓ COX-2↓ AQP-3↑		[110]
Oleanolic acid	PM	Skin aging	AhR	CYP1A1 mRNA↓ TNF-α mRNA↓ IL-6 secretion↓ MMP-1 expression↓ P62, LC-3 I, II↓		[111]

**Table 4 ijms-24-05953-t004:** Experimental mouse models of AhR-related skin diseases.

Stimuli	Method	Target Disease	In Vivo Markers	Ref
Calcipotriol (MC903)	AhR-null mice	Atopic dermatitis	Ear thickness Scratching frequency	[113]
MC903	AhR antagonist (CH223191)	Atopic dermatitis	Gross appearance of mice ears Ear thickness Scratching frequency TSLP mRNA expression Serum IgE level	[113]
IMQ	AhR+/+, AhR −/− mice	Psoriasis Atopic dermatitis	Clinical score IL-17A, IL-17F secretion	[114]
LL-37	AhR agonist (benvitimod)	Rosacea-like eruptions	Redness score Redness area Dermal inflammatory cell infiltration TLR2 expression Chemokine expression	[115]
Bleomycin (BLM)	AhR antagonist (CH223191)	Skin fibrosis	Dermal thickness COL1A1, α-SMA, CTGF, TGF-β1 mRNA level	[116]
	AhR-KO mice	Skin barrier integrity	Trans-epidermal water loss (TEWL) Structural changes in epidermis Microarray	[117]
	Removal of AhR ligands from diet	Skin barrier integrity	Barrier integrity	[117]
	Constitutive active form of AhR (AhR-CA) transgenic mice	Skin inflammation	Total duration of grooming behavior Scratching frequency Severity of skin disorder CYP1A1 expression CYP1B1, NQO1, ADH, AhRR, K1, K6, K16, CCL20, IL-18 gene expression IgG1, IgE, IL-4, IL-5 secretion	[118]

## Data Availability

The data presented in this study are available.

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
