# Peer review of "Preventive Effect of Pharmaceutical Phytochemicals Targeting the Src Family of Protein Tyrosine Kinases and Aryl Hydrocarbon Receptor on Environmental Stress-Induced Skin Disease"

_ijms, 2023, doi:10.3390/ijms24065953_

Round 1
Reviewer 1 Report
This manuscript titled “Preventive effect of pharmaceutical phytochemicals targeting the Src family of protein tyrosine kinases and AhR on environmental stress-induced skin disease”. The comments for this manuscript are as follows:
1. All scientific names should be italicized, this is common sense, please check the full manuscriptt carefully and correct it. For example Gelidium amansii etc in Ref. 5,6.
2. There are still many errors in the section of "Refererences". The writing of references should be consistent, and each word should not be capitalized. Please according to the "Instructions for Authors" to rewrite the references. For example Ref. 14, 22, 27, 40, 45, 46, 50, 56 etc…
3. Although this review manuscript has collected a lot of references, the reviewer think that it does not have the level to be a review paper. However his structure, description or conclusion of the manuscript, the reviewer believes that this review manuscript is only little help for readers. In the current situation, it is not recommended to publish it in the "International Journal of Molecular Sciences". In short, this review manuscript should be substantially revised, and should have more specific conclusions, otherwise it looks like a hodgepodge manuscript. For example, the authors have better suggestions and feasible methods.
I decided it should be a major revision.
Author Response
Please see the attachment.
Point 1: All scientific names should be italicized, this is common sense, please check the full manuscript carefully and correct it. For example Gelidium amansii etc in Ref. 5,6.
Response 1: Thank you for your comment. We have made the appropriate corrections.
Point 2: There are still many errors in the section of "Refererences". The writing of references should be consistent, and each word should not be capitalized. Please according to the "Instructions for Authors" to rewrite the references. For example Ref. 14, 22, 27, 40, 45, 46, 50, 56 etc…
Response 2: Thank you for your comment. We have made the appropriate corrections.
Point 3: Although this review manuscript has collected a lot of references, the reviewer think that it does not have the level to be a review paper. However his structure, description or conclusion of the manuscript, the reviewer believes that this review manuscript is only little help for readers. In the current situation, it is not recommended to publish it in the "International Journal of Molecular Sciences". In short, this review manuscript should be substantially revised, and should have more specific conclusions, otherwise it looks like a hodgepodge manuscript. For example, the authors have better suggestions and feasible methods.
Response 3: To address the reviewer's points, we have revised the overall structure of the manuscript and clarified the purpose of our study. In this paper, we aimed to investigate cases in which phytochemicals directly bind to SFK and AhR that have been overstimulated by environmental factors to regulate these activities and ultimately prevent skin diseases. In these cases, we attempted to describe the molecular mechanism in detail, and the papers that examined signal transduction, rather than relatively direct effects, mentioned the upper and lower signals of SFK and AhR and described the related action principle, which will allow the identification of various skin diseases and other kinases in the future. Therefore, it is expected to be useful for targeting future areas of research. In addition, this study is thought to provide useful data to guide research and development of nutraceutical or pharmaceutical phytochemicals for controlling skin diseases with SFK and AhR as the main targets.

Reviewer 2 Report
The effort made by the authors is very valuable, as nowadays it is impossible to follow most of the current literature on a topic of interest. This paper is a comprehensive review focused on the potential molecular target of the Src family of protein tyrosine kinases and AhR and their target phytochemicals in environmental stress-induced skin diseases. The manuscript fits within the scope of the journal. The manuscript is interesting and the idea is nice. The title is clear and adequate for the article’s content. The author’s work on discussing achieved results is appreciated. The revisions are necessary to improve the clarity of the presentation.
I have some recommendations for authors:
Are the images original? If yes please include the software used.
Include a subchapter in which you describe the working method.
Please include more future research directions.
Author Response
Please see the attachment.
Point 1: I have some recommendations for authors:
Response 1: Thank you for this suggestion.
Point 2: Are the images original? If yes please include the software used.
Response 2: Thank you for your comment. All images were newly created by authors using Microsoft® PowerPoint™.
Point 3: Include a subchapter in which you describe the working method.
Response 3: Thank you for your comment. We have made the suggested changes.
Point 4: Please include more future research directions.
Response 4: Thank you for your comment. We have included additional information related to future research directions in the revised manuscript.

Round 2
Reviewer 1 Report
The authors of this review manuscript has been greatly revised according to the reviewer's suggestions, and I think it can be accepted and published by . Int J Med Sci.